# Non-Supported Nickel-Based Coral Sponge-Like Porous Magnetic Alloys for Catalytic Production of Syngas and Carbon Bio-Nanofilaments via a Biogas Decomposition Approach

**DOI:** 10.3390/nano8121053

**Published:** 2018-12-14

**Authors:** Buthainah Ali, Siti Masrinda Tasirin, Payam Aminayi, Zahira Yaakob, Nur Tantiyani Ali, Wadhah Noori

**Affiliations:** 1Research Center for Sustainable Process Technology (CESPRO), Faculty of Engineering and Built Environment, Universiti Kebangsaan Malaysia, UKM Bangi 43600, Selangor, Malaysia; zahirayaakob65@gmail.com (Z.Y.); tantiyani@ukm.edu.my (N.T.A.); 2Chemical Engineering Programme, Faculty of Engineering and Built Environment, Universiti Kebangsaan Malaysia, UKM Bangi 43600, Selangor, Malaysia; 3Department of Management, Terry College of Business, University of Georgia, Athens, GA 30602, USA; payam.aminayi@gmail.com; 4Department of Biological and Agricultural Engineering, Faculty of Engineering, University Putra Malaysia, Serdang 43400, Selangor, Malaysia; wadah.noori@yahoo.com

**Keywords:** biogas decomposition, unsupported catalysts, magnetic alloys, coral sponge, Syngas, carbon bio-nanofilaments

## Abstract

Porous Ni, Ni-Co, Ni-Fe, and Ni-Cu magnetic alloys with a morphology similar to a giant barrel sponge were synthesized via a facile co-precipitation procedure and then by hydrogen reduction treatment. For the first time, the non-supported alloys with their unique morphology were employed in catalytic biogas decomposition (CBD) at a reaction temperature of 700 °C and 100 mL min^−1^ to produce syngas and carbon bio-nanofilaments, and the catalysts’ behavior, CH_4_ and CO_2_ conversion, and the carbon produced during the reaction were investigated. All of the equimolar alloy catalysts showed good activity and stability for the catalytic biogas decomposition. The highest sustainability factor (0.66) and carbon yield (424%) were accomplished with the Ni-Co alloy without any significant inactivation for six hours, while the highest carbon efficiency of 36.43 was obtained with the Ni-Co catalyst, which is considered relatively low in comparison with industry standards, indicating a low carbon production process efficiency, possibly due to the relatively high biogas flow rate. The higher activity of the Ni-Co alloy catalyst was associated with the synergistic impact between nickel and cobalt, allowing the catalyst to maintain a high stability throughout the reaction period. Moreover, highly uniform, interwoven carbon bio-nanofilaments with a parallel and fishbone structure were achieved.

## 1. Introduction

In the last few years, the resources of renewable energy, in particular biogas, have gained massive attention around the world as a substitute for traditional fossil fuels. Biogas is obtained from the process of the anaerobic digestion of organic compounds [1]. Methane (40–70%) and carbon dioxide (30–60%) are the primary compounds of biogas and can be utilized for heat generation, electricity production, the production of bio-methane, and the production of synthesis gas (referred to as syngas, mixture of H_2_ and CO) [2]. The dry reforming of methane (DRM) is a process through which syngas can be obtained from biogas with the following reaction:CH_4_ + CO_2_ → 2H_2_+ 2CO, ∆H°_298K_ = 247 kJ mol^−1^(1)

This process has become the most attractive process to convert two greenhouse gases of CH_4_ and CO_2_ into synthesis gas (H_2_ and CO). However, the DRM process can be problematic as catalyst deactivation can occur due to the carbon deposition during the decomposition of methane and the disproportionation reactions of carbon dioxide [3]. The carbon formation problem is aggravated when biogas is utilized for such a process, as the CH_4_:CO_2_ molar ratio of biogas is greater than one which could lead to a significant formation of carbon deposits in a short time [4]. Pinilla et al. [5] reported that when considering catalyst activity, the carbon atoms type and location are more important than the amount of carbon produced. In general, the encapsulating carbon is the primary reason for catalyst deactivation, because it directly deposits on the active sites of the catalysts rather than the carrier surface. Other forms of carbon, such as nanofilaments, are not found to be responsible for catalyst deactivation; rather, when generated in a significant amount, they can only cause a reactor blockage.

Carbon nanofilaments, including nanofibers and nanotubes, are formed when stacks of graphene layers are rolled into filaments with diameters ranging from (1–200) nanometers and lengths of several micrometers. The orientation of the carbons in the graphene structure can be used to classify the carbon nanofilaments [6]. Carbon nanofilaments possess exceptional properties, such as high thermal and electrical conductivity, and unique morphological properties that make them useful in the areas of catalysis [7], magnetic materials [8], or as an additive in polymer composites [9]. The synthesis of these substances from an inexhaustible source, for example, biogas, rather than other non-sustainable feedstocks ordinarily utilized would permit them to be categorised as bio-based materials.

Exploiting the feature of the potential of getting bio-based materials from biogas, Pinilla et al.’s research group have recently proposed the catalytic decomposition of biogas to produce carbon nanofilaments from biogas, as well as synthesis gas, in a typical ratio close to 1 [10]. Some previous works by Llobet et al. [11,12] focused on a study of the process variables, such as temperature, space velocity, and CH_4_:CO_2_ partial pressures. As per their report, the Ni/Al_2_O_3_ catalyst exhibited high activity as well as stability, allowing them to obtain high CH_4_ conversion together with the high-yield production of fishbone-like nanocarbon. Other active catalysts such as Fe and Co were also reported for the catalytic decomposition of biogas. However, the activity and stability of Fe and Co were reported to be significantly lower than Ni-based catalysts due to the formation of encapsulating carbon.

In the past few years, a great number of efforts have been made by researchers to synthesize bimetallic alloys. Transition metals like Ni, Co, Fe, and Cu have attracted interest because they are inexpensive and have a high magnetization capacity [13]. Furthermore, bimetallic alloys such as Ni-Co, Ni-Fe, and Ni-Cu have become very attractive to researchers due to their properties and the diversity of applications when compared with their individual mono-metal counterparts. Additionally, the incorporation of nickel into Co, Fe, and Cu metals decreases the use of expensive noble metals [14].

Recently, many attempts have been made to utilize bimetallic catalysts in the catalytic process [15,16]. The success of bimetallic catalysts is believed to be because they consist of two distinct metals that exhibit high dispersion and active sites owing to the synergy of their parent metals. Moreover, the physical and chemical properties of the bimetallic catalysts are enhanced due to the formation of the solid solution [7,17,18]. For instance, Pudukudy et al. [19] and Pinilla et al. [20] reported that a bimetallic catalyst shows a higher carbon yield in comparison with a monometallic catalyst. Bimetallic alloys have been synthesized using various physical and chemical techniques, including sol–gel [21], hydrothermal [22], pyrolysis [23], sonochemical [24], radiolysis [25], microwave combustion [26], microemulsion [27], impregnation [28], and precipitation [29]. However, investigation of the synthesis of bimetallic magnetic alloys is still attractive to researchers [30,31,32,33].

Since the primary goal of the biogas decomposition approach is the production of syngas, for the first time in this study, our focus was on the catalytic activity of non-supported coral sponge porous Ni, Ni-Co, Ni-Fe, and Ni-Cu magnetic alloys synthesized via a facile co-precipitation procedure and then by a hydrogen reduction treatment. The prepared mixed oxides and alloys were characterized for their morphological, crystallography, redox, and magnetic properties. The alloys were utilized in the catalytic biogas decomposition (CBD) reaction and the catalytic activity in the conversion of CH_4_ and CO_2_ was analyzed. In addition, the carbon bio-nanofilaments obtained after catalytic biogas decomposition were analyzed using analytical chemistry methods. As far as we know, this paper is the first to study non-supported porous coral sponge-like Ni, Ni-Co, Ni-Fe, and Ni-Cu magnetic alloys for catalytic biogas decomposition.

## 2. Materials and Methods

### 2.1. Synthesis of Bimetallic Alloys

In this study, analytical grade Ni(NO_3_)_2_·6H_2_O (R&M Chemicals, Subang, Selangor, Malaysia), Co(NO_3_)_2_·6H_2_O (R&M Chemicals, Subang, Selangor, Malaysia), Fe(NO_3_)_3_·9H_2_O (R&M Chemicals, Subang, Selangor, Malaysia), and Cu(NO_3_)_2_·3H_2_O (R&M Chemicals, Subang, Selangor, Malaysia) were utilized as precursors without further purification. (NH_4_)_2_CO_3_ (R&M Chemicals, Subang, Selangor, Malaysia) and C_6_H_8_O_7_·H_2_O (Merck, Shah Alam, Selangor, Malaysia) were utilized as precipitating agents. Bimetallic alloys were synthesized via a facile co-precipitation procedure and then by a hydrogen reduction treatment. Firstly, the mixture of Ni(NO_3_)_2_·6H_2_O (0.1 mol) and M (M = Co(NO_3_)_2_·6H_2_O, Fe(NO_3_)_3_·9H_2_O, Cu(NO_3_)_2_·3H_2_O) (0.1 mol) was thawed in one litter beaker containing five hundred milliliters of deionized water. The mixture was kept under constant agitation at 25 °C for 15 min. With the intention to improve the porosity in the final product, (NH_4_)_2_CO_3_ (0.2 mol) was then added into the salt mixture. The solution was kept under constant agitation until the metal salt and precipitating agent were completely dissolved. In order to control the pH in the final solution at 11, citric acid monohydrate (C_6_H_8_O_7_·H_2_O) was added dropwise to the solution. Subsequently, under agitation using a magnetic stirrer, the final solution was stirred for another one hour at 60 °C to complete the reaction following the completion of citric acid addition. After that, the precipitants were cooled down to 25 °C and then laundered several times using hot deionized water. Then, the samples were centrifuged and dried for 12 h at 100 °C using an air oven. Thereafter, the dried precipitants were smashed to fine powder and calcined for five hours at 600 °C to remove the carbon dioxide from the samples and to obtain the mixed oxides. A portion of the prepared mixed oxide samples was collected for further characterization (discussed in Section 3.1).

Finally, the bimetallic alloys were obtained via the hydrogen reduction treatment. The reduction of the as-synthesized mixed oxides was accomplished in a tubular fixed bed reactor under atmospheric pressure, and a continuous hydrogen flow (150 mL min^−1^) for 120 min at 700 °C (the temperature was experimentally identified using TPR analysis). After the reduction treatment, the reactor was flushed with a flow of nitrogen (50 mL min^−1^) until settled down to 25 °C and the reduced samples were collected for further characterization (discussed in Section 3.2).

### 2.2. Catalytic Biogas Decomposition Procedure

To conduct the experiments, a methane and carbon dioxide mixture of 60:40 *v*/*v* (typical CH_4_:CO_2_ of biogas) was employed. A catalytic biogas decomposition reaction was accomplished in a tubular fixed bed reactor (60 cm L, 3.1 cm O.D, 2.5 cm I.D) with three compartments made of stainless steel under atmospheric pressure, heated up using an electric furnace. In the middle chamber, wrapped in thermal resistive quartz wool, 0.8 g of alloys sample was loosely packed. Initially, the reactor was flushed with a nitrogen gas (150 mL min^−1^) for 60 min to eliminate any undesired gases and to increase the temperature of the reactor from 25 °C to 700 °C. The catalytic reactions were performed at 700 °C with 60% CH_4_:40% CO_2_ synthetic gas (AGS Gases, Selangor, Malaysia) at the flow rate of 100 mL min^−1^ for six hours under atmospheric pressure. After the reaction cycle, the reactor was flushed with a flow of nitrogen (50 mL min^−1^) until settled down to 25 °C to collect the carbon nanofilaments for further characterization.

Using the calibration data, methane conversion ($X_{{\mathrm{CH}}_{4}}$), carbon dioxide conversion ($X_{{\mathrm{CO}}_{2}}$), reaction rate ($-r_{{\mathrm{CH}}_{4}}$), CH_4_ sustainability factor (${S.F}_{{\mathrm{CH}}_{4}}$), hydrogen yield (H_2_), and carbon monoxide yield (CO) were calculated and used to obtain the H_2_:CO ratio (hydrogen yield/carbon monoxide yield) [16,34].

The outlet syngas composition was established by collecting bags of samples at fixed time intervals. Agilent Technologies, 7890A gas chromatography equipped with one column (Carboxen-1010PLOTCapillary) connected to the TCD detector with argon as a carrier gas (99.999%, AGS Gases, Selangor, Malaysia) was used to analyze the collected outlet gases (H_2_, CO, CH_4_, and CO_2_) and to detect their concentrations.

### 2.3. Characterization of the Freshly Oxides, Alloys and Employed Catalysts

The crystallography and phase purity of the fresh oxides, alloys, and employed catalysts were determined by the technique of X-ray diffraction phase (XRD) (Bruker D8 Advance, Bruker, Billerica, MA, USA) with a Cu Kα radiation wavelength of 1.54 Å as the X-ray source.

The temperature-programmed reduction (TPR) technique (Micromeritics Autochem 2920 chemisorption analyzer, Micromeritics Instrument Corp., Norcross, GA, USA) was employed to examine the reducibility of the samples before reduction. The analysis was accomplished from 25 °C to 800 °C with a heating rate of 10 °C per minute (in case of Ni-Fe, the temperature was increased to 1000 °C to cover all important reduction peaks), under a 10% hydrogen/nitrogen gas mixture flow (20 mL min^−1^).

The nitrogen adsorption/desorption of the samples before and after the reduction was investigated by a Micromeritics 3FLEX surface characterization analyzer (Micromeritics Instrument Corp., Norcross, GA, USA). To eliminate the adsorbed impurities, each sample was minutely degassed at 300 °C for 6 h prior to the measurements being taken. The Brunauer-Emmett-Teller method (BET) was employed to measure the surface area, whereas the Barrett-Joyner-Halenda method (BJH) was used to measure the samples’ pore size characteristics.

The morphological appearance of the fresh oxide and alloy samples was visualized using a high resolution Zeiss Merlin field emission scanning electron microscope (FE-SEM, Oxford Instruments Pte. Ltd., Ubi Techpark Lobby E, Singapore) with an operating voltage of 3 kV. The elemental composition of the reduced samples was analyzed using an electron dispersive X-ray (EDX, Oxford Instruments Pte. Ltd., Ubi Techpark Lobby E, Singapore) instrument, fitted with the FE-SEM equipment. The morphology of the produced carbon bio-nanofilaments was investigated by transmission electron microscopy (TEM) (Philips CM-12, Crest Systems (M) Sdn Bhd, One Puchong Business Park, Puchong, Selangor, Malaysia) equipment, controlled at an accelerating voltage of 100 kV.

The saturation magnetization and magnetic hysteresis loops of the reduced samples were observed using a vibrating sample magnetometer (VSM) (LakeShore magnetometer Lake Shore Cryotronics, Inc., 575 McCorkle Blvd., Westerville, OH, USA) run at 25 °C with a range of magnetic field from +14 KOe to −14 KOe.

The thermogravimetric (TGA)/differential thermal (DTA) technique (Mettler Toledo SDTA851e, Mettler-Toledo (M) Sdn Bhd, Bukit Jelutong, Shah Alam, Selangor, Malaysia) was employed to evaluated the thermal stability, purity, yield, and efficiency of the produced carbon. The carbonous materials were heated up from 25 °C to 900 °C in the presence of air, at a heating rate of 10 °C min^−1^. For each catalyst, the carbonous yield (Y_C_) was calculated according to Equation (2) [35] and the Ni, Co, Fe, and Cu oxidation was taken into account to make calculations:(2)$$Y_{C}\left(\%\right)=\frac{\%\;Weight\;loss\;by\;carbon\;oxidation\;}{\%\;of\;residue\;after\;oxidation}\times 100$$

The carbon efficiency (C_E_) is the ratio between the carbon deposited amount determined from thermogravimetric (TGA) results in mole and the methane (CH_4_) total amount shown in mole converted during six hours of reaction [20].

Raman spectroscopic analysis was utilized to study the crystalline and graphitization degree of the carbonous materials. All measurements were conducted at 25 °C by a Raman spectrometer (WITec500588107, WITec Pte. Ltd., International Business Park, Germen Center, Singapore) fitted with a diode Nd: YAG laser, as well as a 532 nm wavelength from 10 to 4000 cm^−1^.

## 3. Results and Discussion

### 3.1. Characterization of the Fresh Oxides

Figure 1 shows the crystallography diffraction patterns of the fresh oxides synthesized by the co-precipitation of the nitrate solutions of Ni, Co, Fe, and Cu metals for the samples obtained prior to reduction.

In Figure 1a, the sharp peaks situated at the 2θ values of 37.3°, 43.3°, 62.9°, 75.5°, and 79.5° are entitled to the face-centered cubic phase and confirm the high crystallinity and phase purity of NiO (JCPDS: 00-047-1049). The X-ray diffraction patterns of Ni-Co mixed oxide are presented in Figure 1b. The peaks situated at 18.9°, 31.1°, 65.2°, 77.0°, and 79.3° can be ascribed to spinel Co_3_O_4_ (JCPDS: 01-073-1701). In addition, the peaks at 37.3°, 43.3°, 62.9°, and 75.5° could be assigned to the NiO crystals and diffraction peaks at 38.4°, 44.8°, 55.4°, 59.4°, 74.1°, and 75.3° could be attributed to NiCo_2_O_4_ crystals. The crystallographic structure of the Ni-Co mixed oxide was found to be face-centered cubic. The Ni-Fe mixed oxide diffraction pattern showed sharp peaks at the 2θ values of 24.3°, 33.3°, 40.9°, 49.6°, and 64.2°, as shown in Figure 1c, corresponding to the presence of the Rhombohedral phase crystallographic structure of Fe_2_O_3_ (JCPDS: 01-077-9926). The intensive peaks situated at the 2θ values of 18.5°, 30.4°, 35.8°, 37.4°, 43.5°, 54.2°, 57.5°, 63°, and 74.6° are ascribed to the face-centered cubic phase crystallographic structure of spinel NiFe_2_O_4_ (JCPDS: 01-076-6118). Moreover, the existence of the sharp diffraction peaks at 2θ at 69.7°, 72.1°, and 75.6° indicates the face-centered cubic crystallographic structure of FeNi_2_O_4_ (JCPDS: 01-074-6507). Figure 1d shows the XRD diffraction pattern of Ni-Cu mixed oxide. The peaks situated at 2θ values of 32.5°, 35.6°, 38.7°, 46.4°, 48.9°, 51.4°, 53.5°, 58.3°, 61.6°, 66.3°, 68.0°, and 72.5° can be assigned to the monoclinic crystallographic structure of CuO (JCPDS: 01-073-6023). The sharp peaks situated at 2θ values of 37.3°, 43.3°, 62.9°, 75.5°, and 79.5° confirm the phase of NiO and NiCuO solid solution (JCPDS: 00-047-1049) and (JCPDS data: 01-078-0645). Scherrer’s equation was used based on the XRD patterns and the average crystallographic size of the fresh oxides was calculated to be 39 nm, 35 nm, 34 nm, and 33 nm for the Ni, Ni-Co, Ni-Fe, and Ni-Cu mixed oxides, respectively. As observed in Figure 1, the XRD patterns of the synthesized Ni, Ni-Co, Ni-Fe, and Ni-Cu mixed oxides show sharp peaks with high intensity, indicating the high crystallinity of the formed oxides [36].

The reducibility of the fresh oxides was studied using the TPR technique in order to experimentally identify the reduction temperatures [20]. The reduction profiles of the synthesized fresh oxides are shown in Figure 2.

The fresh oxides displayed a strong reduction peak <500 °C and weak reduction peak >500 °C. These peaks are attributed to Ni, Co, Fe, and Cu reduction in oxide forms (NiO, Co_3_O_4_, NiCo_2_O_4_, Fe_2_O_3_, NiFe_2_O_4_, FeNi_2_O_4_, CuO, NiCuO). Indeed, the presence of the mentioned oxides was revealed using X-ray diffraction analysis. As shown in Figure 2, the TPR profile of NiO displays two resolved reduction peaks at 377 °C and 523 °C. These two peaks are ascribed to the reduction steps of Ni^2+^→Ni. As presented in Figure 2, the bimetallic Ni-Co oxide profile shows a strong peak situated at 380 °C and a weak peak centered at 557 °C, corresponding to the reduction of Ni^2+^→Ni and Co^3+^→Co^2+^→Co, respectively. The Ni-Fe mixed oxide profile presents three reduction peaks (Figure 2), indicating the reduction of metal oxides into their metallic states. The first peak situated at 392 °C corresponds to the reduction of Fe^3+^ to Fe^2+^, in Fe_2_O_3_. The reduction shoulder at 624 °C is ascribed to the reduction of NiO from valence Ni^2+^ to metallic Ni, and Fe_3_O_4_ from Fe^3+^ to Fe^2+^. In addition, the third peak at 919 °C is attributed to the reduction of NiFe_2_O_4_, FeNi_2_O_4_, and FeO. Hence, the reduction profile of bimetallic Ni-Fe indicates the reduction steps of NiO→Ni and Fe_2_O_3_→Fe_3_O_4_→FeO→Fe [37]. The TPR profile of the Ni-Cu mixed oxide also presents two reduction peaks at a lower temperature than other mixed oxides or the Ni oxide alone (Figure 2). The broad peak at 266 °C, along with a small shoulder at 330 °C, is ascribed to the reduction of NiO and CuO to metallic Ni or Cu, respectively, and the second peak observed at 553 °C is attributed to the reduction of NiCuO solid solution. The two resolved reduction peaks could be attributed to the sequential step reduction of Ni^2+^ and Cu^2+^. The shift in the reduction temperature to a lower temperature was reported to be because of the rapid CuO reduction, and the copper existence could greatly improve the Ni species reducibility [38]. In addition, another study [39] has proposed that Cu can dissociate the molecule of hydrogen into hydrogen atoms which can spill over to the neighboring NiO surface and lead to a lower reduction temperature of NiO in comparison with that of the pure Ni. As such, the TPR profile of the Ni-Cu mixed oxide reveals that Ni is well alloyed with Cu in bimetallic Ni-Cu samples. This observation is harmonious with preceding research findings [38,39] that nickel can shape a homogeneous combination with copper in any atomic proportion. Based on the TPR spectra of Ni-Co and Ni-Fe mixed oxides, it can be noted that the crystal structure of nickel oxide effectively activates Co^3+^ and Fe^3+^ ions. The reason for this is that Co and Fe atoms, as doping species, are doped into a nickel oxide crystal structure. Given that Co and Fe atoms have higher valences than nickel atoms, the cobalt and iron dangle bonds increase, which makes Co^3+^ and Fe^3+^ more effective [40]. For all prepared samples, the theoretical amount of hydrogen, calculated from mass balance analysis, and the experimental amount of hydrogen consumed during the course of the TPR measurement, are presented in Table 1.

It can be noted that there is a relatively good agreement between the theoretical and the experimental values for the hydrogen consumption. Therefore, it can be concluded that all the TPR peaks shown in Figure 2 are most likely only related to the reduction of Ni, Ni-Co, Ni-Fe, and Ni-Cu species [16,41].

Textural properties such as specific surface area, pore volume, and pore diameter of the fresh oxide samples were studied utilizing the BET/BJH method, and the results are tabulated in Table 2.

According to IUPAC classification, the isotherms (Figure 3) of the mixed oxides were classified as type IV with an H3 hysteresis loop, suggesting the existence of mesoporous materials with an incision-like pore geometry. The H3 type hysteresis loops are seen in the non-uniform size-shape aggregated solid nanoparticles. Thus, the pores were predicted to have been created by the inter-aggregation of the metal oxide nanoparticles.

The FE-SEM analysis was utilized to investigate the surface morphology of the as-obtained Ni, Ni-Co, Ni-Fe, and Ni-Cu fresh oxides at different magnifications (Figure 4A–D). At high magnifications, all prepared mixed oxides were noticed to mainly subsist in the shape of semi-spherical nanoparticles. In addition, the network of porous particles with a high surface area can be clearly observed. The freeing of high amounts of carbon oxides from the bulk of the sample during the calcination process may have been accountable for the creation of this clearly porous texture of the fresh oxides.

### 3.2. Characterization of Bimetallic Alloys

The crystallography of the reduced samples was characterized using the X-ray diffraction technique and the reduction patterns are shown in Figure 5.

In Figure 5a, the sharp peaks centered at the 2θ values of 44.7°, 52.0°, and 76.5° are designated to the face-centered cubic phase crystallographic of metallic Ni, formed by the reduction of NiO crystals (JCPDS data: 01-071-4655). The X-ray diffraction pattern of the reduced Ni-Co mixed oxide is presented in Figure 5b. The peaks situated at 44.4°, 52°, and 76.2° are ascribed to the formation of the face-centered cubic phase crystallographic structure of the Ni-Co alloy (JCPDS data: 01-074-5694). No other pattern peaks had been present in the spectra, showing the high crystallinity and phase purity of the reduced Ni-Co alloy sample. Figure 5c shows the intensive pattern peaks situated at the 2θ values of 44.7° and 65° ascribed to the formation of the face-centered cubic phase crystallographic structure of the Ni-Fe bimetallic alloy (JCPDS data: 01-071-8326). Additionally, the weak pattern peaks situated at the 2θ values of 51°, 57°, and 75° are ascribed to the monoclinic phase crystallographic structure of the Ni-Fe alloy (JCPDS data: 01-071-5097). On the other hand, the pattern peaks situated at the 2θ values 30.1°, 35.5°, and 62.7° (Figure 5c) are attributed to the cubic phase crystallographic structure of Fe_3_O_4_ (JCPDS: 01-086-1344). Figure 5d presents the diffraction patterns of the reduced NiCu mixed oxide. The diffraction peaks located at the 2θ values of 43.8°, 51°, and 75° may be assigned to the face-centered cubic phase crystallographic structure of the Ni-Cu alloy (JCPDS data: 00-047-1406). No other diffraction peaks were identified in the XRD spectra, denoting the high crystallinity and phase purity of the bimetallic alloy. The high intensity of the pattern peaks determines the great crystalline nature of the formed alloys [42,43,44,45,46].

Textural properties (specific surface area, pore volume, and pore diameter) of the bimetallic alloys were studied utilizing the BET/BJH method, and the results are tabulated in Table 2. As depicted in Figure 6, the nitrogen isotherms of the Ni, Ni-Co, Ni-Fe, and Ni-Cu alloys were classified as type IV with an H3 hysteresis loop, suggesting the formation of mesoporous alloys with an incision-like pore geometry, similar to the mixed oxide results. From Table 2, it is clear that the parameters such as specific surface area, pore volume, and pore size of the mixed oxides were reduced after the reduction treatment at 700 °C, which further points to the formation of alloys. This is primarily due to the removal of water during dehydration, which results in the inter-aggregation of particles and, subsequently, a decrease in the specific surface area and pore size of the particles [47,48,49].

The energy dispersive spectroscopy technique (EDX) was used to study the quantitative and qualitative composition of the Ni, Ni-Co, Ni-Fe, and Ni-Cu bimetallic alloys, as shown in Figure 7. The C-peak in all samples arises from the carbon tape material utilized for the FE-SEM/EDX analysis. Figure 7 shows that the reduced Ni sample consisting of only Ni and C (Figure 7a); the Ni-Co alloy consisting of Ni and Co (Figure 7b); the Ni-Fe alloy consisting of Ni, Fe, and O (Figure 7c); and the Ni-Cu alloy consisting of Ni and Cu (Figure 7d) are the primary components. In the Ni-Fe alloy, the peak for O could be identified due to the presence of unreduced nickel ferrites in the sample. No other elemental impurities were identified, indicating the high purity of the synthesized bimetallic alloys.

The high resolution field-emission scanning electron microscopy technique was used to visualize the morphological appearance of the as-obtained Ni, Ni-Co, Ni-Fe, and Ni-Cu alloys. The FE-SEM photographs of the alloys at various magnifications are represented in Figure 8.

As marked in Figure 8, the giant barrel coral sponge-like surface porous morphology was obviously obtained for the nickel-based bimetallic alloy. As noticed in the FE-SEM photographs, the small porous particles of nickel crystallites tend to amalgamate and conglomerate together, resulting in huge porous particle assembles of nickel. Figure 8A shows that the reduced Ni, as well as the prepared bimetallic alloys, has formed a continuous network, likely as a result of the linkage of the nickel porous particles conglomerates, either due to their magnetic characteristics or the removal of water from between the particles, or both. Moreover, the Ni alloy exhibited a highly porous appearance, which might have originated from the freeing of huge amounts of carbon oxides from the bulk of the precipitated intermediate precursor during the calcination process. Furthermore, the body of the Ni alloy was found to be formed from quasi-spherical shape particles with the diameter varying from 10 to 40 nm [50]. Similarly, as depicted in Figure 8B, the Ni-Co alloy exhibits a lobe coral sponge morphology with a high porosity and homogeneity. The size of particles forming the Ni-Co alloy ranges from 20 to 50 nm. It can be noted that the Ni-Co alloy also tends to form continuous porous networks by linkage of the nickel-cobalt particle agglomerates [21]. The Ni-Fe alloy (Figure 8C) shows small lumps at low magnification, whilst at high magnification, the lobe coral sponge morphology and high porous texture with a varying size of 10 to 40 nm can be observed. Similarly, the freeing of carbon oxides from the bulk of the sample during the calcination step might be accountable for the creation of the visible porous texture of the alloy [8]. As can be noted in Figure 8D, the Ni-Cu alloy exhibits a lobe coral sponge-like morphology with a strong agglomeration, with the particle size ranging from 10 to 30 nm [13]. The giant barrel coral sponge-like surface morphology seems to be due to the existence of Ni in bimetallic alloys, irrespective of the second metal. This phenomenon is perhaps because the hydrogen reduction treatment results in the coalescence of the particles into bigger lumps [42,43,44,45,46]. The comparison of the FE-SEM photographs (Figure 4 versus Figure 8) confirms the BET results, where the surface of the particles becomes smoother after the reduction.

The Vibrating Sample Magnetometer (VSM) technique was used to characterize the magnetic properties of the as-obtained Ni, Ni-Co, Ni-Fe, and Ni-Cu bimetallic alloys. The magnetic hysteresis (M − H) of the alloys was recorded in a magnetic field ranging from +14 KOe to −14 KOe at 25 °C. The remanence (Mr), saturation magnetization (Ms), and coercivity (Hc) parameters were extracted from the magnetic hysteresis loop (Figure 9) and presented in Table 3.

The as-obtained Ni, Ni-Co, Ni-Fe, and Ni-Cu bimetallic alloys exhibit S-shape magnetization hysteresis curves corresponding to ferromagnetic soft materials. In comparison with the saturation magnetization (Ms) and coercivity (Hc) values for bulk Ni obtained from the literature (54 emu g^−1^, 6 Oe [51]), the Ms of the prepared metallic Ni is comparable (54.2 emu g^−1^), whereas the coercivity value shows an increase (43.7 Oe). This could be due to the smaller Ni crystal size obtained in this study (39 nm as compared with 33 nm average crystal size obtained in this study) after the reduction process [52]. The saturation magnetization (Ms) of the Ni-Co alloy was found to be 89.5 emu g^−1^, whereas the coercivity (Hc) was found to be 31.9 Oe. Although the crystal size of the Ni-Co alloy (26 nm) is smaller than the single domain size of Ni (33 nm), coercivity showed a decrease in comparison to bulk Ni in the case of the prepared bimetallic Ni-Co alloy [8,53]. The Ni-Fe alloy shows the highest Ms (124.4 emu g^−1^), Mr (10.2 emu g^−1^), and Hc (53.7 Oe) values among the prepared alloys because of its small crystal size (15 nm) and the presence of iron in the alloy [42]. In contrast, the Ni-Cu alloy shows a smaller Ms value (51 emu g^−1^) when compared with the Ms values of Ni, Ni-Co, and Ni-Fe, due to the presence of copper in the alloy. The Ni-Cu alloy shows a higher Hc (46.5 Oe) compared with bulk Ni (43.7 Oe), due to its smaller crystal size (11 nm) [54]. In Table 3, the overall analysis of the saturation magnetization (Ms) and the coercivity (Hc) values seems to suggest an increase with the decrease of crystal size of the alloys. But, it is important to note that this phenomenon may change when a different metal is used [8,52]. Therefore, further investigation is needed to confirm the crystal size dependences of magnetic properties.

### 3.3. Alloys Activity and Stability in the CBD

The biogas decomposition kinetic experiments were carried out over the as-synthesized catalyst alloys of Ni-Co, Ni-Fe, and Ni-Cu (as well as Ni as a reference catalyst) at 700 °C. The flowrate of biogas CH_4_:CO_2_ with a 60:40 (*v*:*v*) ratio was set to 100 mL min^−1^. The results obtained were compared with Ni monometallic catalysts. The catalysts were assessed in terms of their activity and stability (Figure 10). CH_4_ and CO_2_ conversion, the reaction rate, the synthesized H_2_ and CO composition percentage, and the H_2_:CO ratio obtained after 5 and 360 min on stream are tabulated in Table 4.

Methane conversion after 5 min on stream followed this sequence: Ni-Fe > Ni > Ni-Co > Ni-Cu. A similar trend was observed with CO_2_ conversions after 5 min on stream with the following sequence: Ni-Fe > Ni-Co > Ni > Ni-Cu, and showed higher CO_2_ conversions when compared with CH_4_ conversion. It can be noted that the alloy catalysts presented a higher carbon dioxide conversion than the monometallic catalyst.

Regarding the H_2_ and CO concentration of the gas produced after 5 min on stream, it is important to note that the highest H_2_ and CO concentrations were observed with the Ni-Fe and Ni-Co alloys (Table 4). Regarding the H_2_:CO ratio, the synthesis gas produced with the Ni-Fe and Ni-Co alloys showed the highest values of 0.18 and 0.17, respectively, while the lowest H_2_:CO ratios of 1.15 and 1.14 were observed with the Ni monometallic and Ni-Cu alloy, respectively. Carbon formation in the catalytic decomposition of biogas can be associated with a methane decomposition reaction (CH_4_ ↔ 2H_2_ + C, ∆H° = 75 kJ mol^−1^) and carbon gasification reaction (CO_2_ + C ↔ 2CO, ∆H° = 171 kJ mol^−1^) [55]. Hence, carbon potential can be linked to the H_2_:CO ratio. Therefore, Ni-Co, Ni-Fe, and Ni catalysts showed higher carbon formation than the Ni-Cu catalyst at the beginning of the experiment.

The values of methane sustainability factors (${S.F}_{{\mathrm{CH}}_{4}}$) were included in Table 4 as indicators of catalyst stability. As illustrated in Figure 10b, the highest stability was obtained with the Ni-Co (0.66) catalyst, followed by Ni-Fe (0.57), Ni (0.55), and Ni-Cu (0.53) catalysts. Since all methane sustainability factors values have been lower than one, it is able to be deduced that all alloys, underneath the reaction conditions described in this study, suffered from some degree of deactivation during the catalytic biogas decomposition experiments. Alloy stability is also reflected in the $-r_{{\mathrm{CH}}_{4}}$ over time (Figure 10a). As shown in Figure 10a, a decrease of $-r_{{\mathrm{CH}}_{4}}$ with time was noted in all experiments of biogas decomposition. Nevertheless, different deactivation behaviors were observed for each catalyst. Initially, Ni-Co and Ni catalysts showed a similar $-r_{{\mathrm{CH}}_{4}}$ curve in the initial reaction stage, during the first 60 min on stream. Afterward, the Ni catalyst showed a lower $-r_{{\mathrm{CH}}_{4}}$ value in comparison with Ni-Co. The $-r_{{\mathrm{CH}}_{4}}$ value of the Ni-Fe catalyst was the highest amongst all catalysts within the first 60 min of the reaction. After 60 min on stream, the rate of deactivation of Ni-Fe exceeded the Ni-Co catalyst. Finally, Ni-Cu catalyst showed a significant decline of $-r_{{\mathrm{CH}}_{4}}$ during the initial 30 min on a stream of reaction. Finally, the Ni-Cu catalyst showed the lowest sustainability factor value across the board.

Figure 11a shows the H_2_:CO ratio over time. As anticipated from the $-r_{{\mathrm{CH}}_{4}}$ results above, the highest H_2_:CO ratio was obtained with the Ni-Fe (0.18) catalyst, followed by Ni-Co (0.17), Ni (0.15), and Ni-Cu (0.14) catalysts. Indeed, it very well may be seen that the H_2_:CO ratios obtained with the Ni-Fe and Ni-Co catalysts were lower than 1 after 120 and 150 min on stream and after that, their value remained stable, at around 0.8 and 0.9, respectively. On the other hand, during the first 60 min on stream, the H_2_:CO ratio of the Ni and Ni-Cu catalysts remained above one, and then stabilized at around 0.8 and 0.7, respectively.

The yield of carbon deposited over the alloys after six hours on stream of biogas decomposition at 700 °C is presented in Figure 11b and tabulated in Table 4. As observed in Figure 11b, the highest carbon yield of 424% was obtained for the Ni-Co alloy, followed by Ni-Fe (377%), Ni (356%), and Ni-Cu (317%) alloys. It is crucial to observe that the carbon yields of Ni-Co, Ni-Fe, and Ni alloys were significantly higher than the carbon yield obtained with the Ni-Cu alloy. With regards to the carbon efficiency, similar trends to those annotated for the carbon deposited yield were observed. The low carbon yield achieved with the Ni-Cu alloy catalyst could be associated with encapsulating carbon formation, as reported in [56]. It can be concluded that at 700 °C, the Ni-Cu alloy catalyst gradually deactivates because of the high copper amount, which might easily make the particles of the catalyst semi-liquid and debilitate their stability at high temperatures [57]. As a result, encapsulation carbon deposits directly on the active site of the catalyst particles rather than the carrier surface [58]. Ashok et al. [57] have suggested that Cu has a highly affinity with the structure of graphite, which bans the graphite layer formation on the surface of nickel, resulting in a lower carbon layers growth rate on a nickel surface.

Despite the high carbon yield value obtained with most of the catalysts (Table 4), it can be noted that the carbon efficiencies are relatively low, indicating a low process efficiency, mainly due to the relatively high biogas flow rate (100 mL min^−1^) used.

### 3.4. Characterization of Carbon Bio-Nanofilaments

The deposited carbon materials over Ni-Co, Ni-Fe and Ni-Cu bimetallic and Ni monometallic catalysts at 700 °C and 100 mL min^−1^ were characterized using X-ray diffraction, Raman, TGA, and TEM techniques. The crystallography of the deposited carbon samples was characterized utilizing the X-ray diffraction technique, and the patterns are shown in Figure 12. As can be noted, all carbonous materials exhibited an intense (002 plane) diffraction peak at around 2θ = 26°, corresponding to the graphitic carbon on the surface of the monometallic Ni and Ni-Co, Ni-Fe, and Ni-Cu alloy catalysts. It is important to note that no peaks related to Ni, Co, Fe, and Cu oxide form species (NiO, Co_3_O_4_, NiCo_2_O_4_, Fe_2_O_3_, NiFe_2_O_4_, FeNi_2_O_4_, CuO, NiCuO) were detected, indicating that all metals stayed in a reduced phase during the biogas decomposition reaction. According to the XRD patterns, major peaks of alloys and Ni monometallic catalysts were identical amongst all spectra. Nevertheless, considering the Ni-Fe alloy, the weak diffraction peaks appearing at the 2θ values 30.1°, 35.5°, and 62.7° (Figure 12) are attributed to the cubic phase crystalline structure of Fe and Fe_3_C [59]. The presence of these peaks indicates that the magnetite (Fe_3_O_4_) in the alloy catalyst would be reduced to FeO (intermediate oxide) using the hydrogen that was released from the decomposition of CH_4_ during the catalytic decomposition of biogas reaction, following which the FeO transformed to Fe and Fe_3_C (Fe_3_O_4_→FeO→Fe + Fe_3_C) [60]. Some research groups have proposed that the formation of filamentous carbons can be linked to the decomposition of Fe_3_C into Fe and carbon [18,36,60]. Hence, it could be deduced that the catalytic biogas decomposition has resulted in synthesis gas production (Figure 11a).

Table 5 highlights the properties of the obtained carbon structures. In comparison with the Ni monometallic catalyst, the carbon materials obtained with the Ni-Co, Ni-Fe, and Ni-Cu alloys showed slightly high interlayer d-spacing (d_002_) values and a lower crystal domain size (L_C_). The interlayer d-spacing (d_002_) values of alloys are very close to that of the ideal interlayer distance between the two graphite layers (0.3354 nm) [20], indicating that the bimetallic alloys lead to the formation of less graphitic bio-filamentous carbons. Nickel monometallic and Ni-Co, Ni-Fe, and Ni-Cu alloys showed a high crystal domain size in comparison to the reduced value (Table 1). However, the increment in Ni value was more pronounced.

The thermogravimetric analysis (TGA) was utilized to evaluated the thermal stability, purity, and yield of the as-deposited carbonous materials over the Ni, Ni-Co, Ni-Fe, and Ni-Cu alloys. As marked in Figure 13a, the weight loss curves had been expressed by means of the temperature versus the preliminary mass residual percentage. In general, the weight loss is attributed to the carbon in oxygen combustion, thus corresponding to the yield of carbon deposited over the alloys. As observed, all carbonous samples displayed same oxidation trend with only one notable weight loss step. No weight loss was observed underneath the temperature of 400 °C. This observation indicates the non-presence of any amorphous carbon in the carbonous samples as amorphous carbon decomposes underneath 450 °C [61], indicating the high purity of carbon obtained. The thermal stability of the carbon bio-nanofilament materials is affected by bio-nanofilament diameters [62,63], as well as defect sites in graphite walls [64]. The temperature/weight profile, including the preliminary weight loss (onset), the maximum weight loss (inflection), and the final weight loss (end), are shown in Figure 13b.

As illustrated in Table 4, the highest yield was obtained with the Ni-Co alloy compared with the other catalysts. In fact, the carbon nanofilament growth is directly proportional to the catalytic biogas decomposition activity [35]. Compared to other alloys, it is worth observing that the carbon produced with the Ni-Cu alloy exhibits the highest oxidation temperature (623 °C), indicating the high thermal stability of carbon nanofilaments, as well as high graphitization and crystallinity [19].

The Raman spectroscopic technique was utilized to investigate the crystalline and graphitization degree of the produced carbonous materials over the Ni, Ni-Co, Ni-Fe, and Ni-Cu bimetallic alloy catalysts (Figure 14). As observed, all carbonous samples displayed two distinct bands. The first band is well-known as the D band, located at 1349 cm^−1^, and is ascribed to the existence of disorder in the carbon structure, such as imperfections in the carbon structure or amorphous carbon [65]. For the samples used in this study, the D band can be ascribed to the graphitic carbon imperfection samples observed by the thermogravimetric analysis (TGA). The appearance of the D band indicates carbon bio-nanofilaments formation [66]. The second band, well-known as the G band, is centered at 1578 cm^−1^, and is attributed to the graphitized carbon bio-nanofilaments [67,68]. The peak intensity proportion well-known as ID/IG is a beneficial tool utilized to measure the degree of graphitization and crystallography quality of as-obtained carbon bio-nanofilaments [69]. Generally, an ID/IG ratio of less than one is desirable. As can be noted in Table 4, carbon bio-nanofilaments obtained from all catalysts have a high quality and crystallinity, showing peak intensity ratios (ID/IG) of less than 1, except for the Ni-Cu catalyst, for which the ratio was calculated to be 1.06, indicating a slightly lower carbon bio-nanofilaments graphitization degree and the crystallinity. These findings are in agreement with TGA observations.

As marked in Figure 15, both carbon bio-nanofilaments and encapsulating carbon were observed in all carbonous samples. As observed in Figure 15a–d, carbon bio-nanofilaments were produced by the catalytic biogas decomposition over the Ni, Ni-Co, Ni-Fe, and Ni-Cu bimetallic alloy catalysts. At low magnifications, all generated samples were revealed to mostly exist in the form of randomly interwoven carbon nanofilaments with various diameters [67,70,71]. Encapsulating carbons were also clearly observed in the micrographs [72]. The agglomerate black spots exemplify the presence of metallic particles in the samples [34,70].

The high magnification micrographs of the produced samples are presented in Figure 16. The graphitic layers of all samples are visible in these micrographs [73]. Two different types of bio-nanofilaments were observed in all samples: parallel and fishbone [74]. As depicted in Figure 16a,b, the fishbone structure, in which the planes were oriented at an angle to the fiber outer surface (axis) [73], was observed in the sample derived from the nickel monometallic catalyst and Ni-Co alloy, along with a small amount of encapsulating carbon covering the metallic particles. In TEM micrographs, carbon fibers with a parallel structure can also be noted. According to de Llobet et al. [34], it is probable that the presence of two structures within the same sample is due to the fact the graphene layers’ inclination relies upon the angle formed by the walls of the particles, as well as the varying of the particle shape. A similar observation can be noted in Figure 16c,d for the carbon bio-nanofilaments produced over the Ni-Fe and Ni-Cu catalysts.

## 4. Conclusions

In summary, magnetic coral sponge-like porous Ni, Ni-Co, Ni-Fe, and Ni-Cu bimetallic alloys were synthesized via a facile double pot method, including co-precipitation followed by a hydrogen reduction treatment method. The non-supported alloys were utilized for the direct decomposition of biogas, composed of CH_4_ and CO_2_, in order to produce syngas, containing H_2_ and CO in a ratio close to 1, in conjunction with carbon bio-nanofilaments. Various characterization techniques were utilized to obtain the crystalline, structural, redox, textural, morphological, and magnetic properties of the fresh oxides and formed bimetallic alloys. The face-centered cubic phase of Ni, Ni-Co, Ni-Fe, and Ni-Cu bimetallic alloys with high crystallinity was found to be formed by the reduction of their representative oxides, as revealed by the XRD analysis. The scanning electron microscopy images were employed to visualize the highly porous giant barrel sponge-like surface morphology of the alloys, and the BET analysis was utilized to examine the textural parameters of the mixed oxides, as well as the bimetallic alloys. The magnetic hysteresis loop measurement of alloys seems to suggest that the saturation magnetization (Ms) and the coercivity (Hc) increase with the decrease in the size of the alloys’ crystals. The magnetic properties of the alloys are also shown to rely on the metal type. All of the alloy catalysts with an equimolar Ni-M (M = Co, Fe, Cu) composition presented good activity and stability for the catalytic biogas decomposition. The highest sustainability factor (0.66) and carbon yield (424%) values were achieved with the Ni-Co alloy at 700 °C and 100 mL min^−1^ without a significant inactivation for six hours, while a high carbon yield efficiency of 36.43 was obtained with the Ni-Co catalyst; this value is relatively low compared to the industry economical value, mainly due to the relatively high biogas flow rate (100 mL min^−1^) used. A lower biogas flow rate would be more conducive to the formation of carbon in the CBD process. The superior activity of the Ni-Co alloy catalyst was associated with the synergistic impact of nickel and cobalt—the Ni presence favored the diffusion of the surface carbon, whereas the Co presence favored the bloat and oxidation of carbon particles via oxidation-reduction reactions. The combination of both effects allows the Ni-Co alloy catalyst to produce a significant yield of carbon bio-nanofilaments by preserving the stability of the catalyst. Highly uniform, interwoven carbon nanofibers with parallel and fishbone structures were obtained, showing a high crystallinity, oxidation stability, and graphitization degree (ID/IG = 0.82) with all of the catalysts.

## Figures and Tables

**Figure 1 nanomaterials-08-01053-f001:**
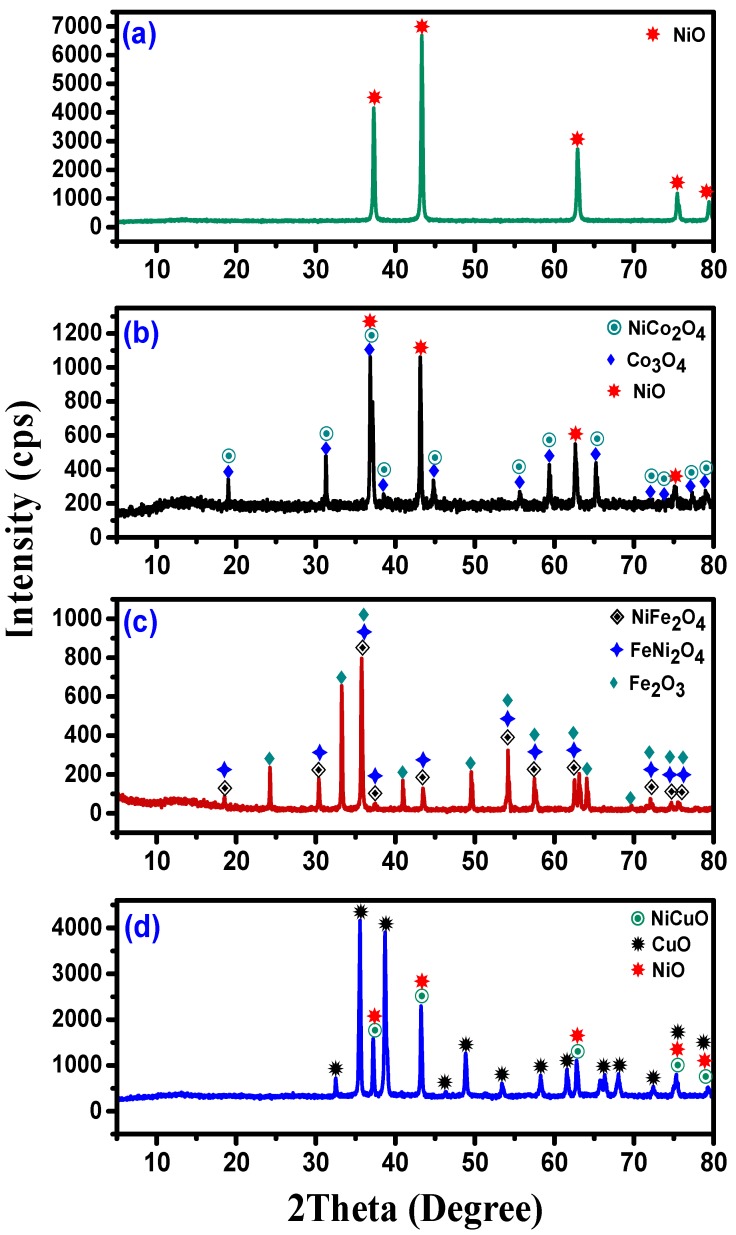
XRD patterns of the fresh samples: (**a**) Ni, (**b**) Ni-Co, (**c**) Ni-Fe, and (**d**) Ni-Cu.

**Figure 2 nanomaterials-08-01053-f002:**
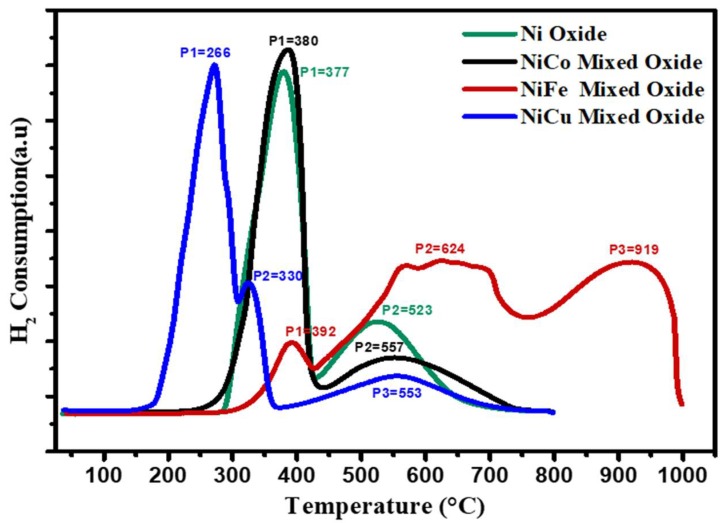
TPR profiles of the fresh samples Ni, Ni-Co, Ni-Fe, and Ni-Cu.

**Figure 3 nanomaterials-08-01053-f003:**
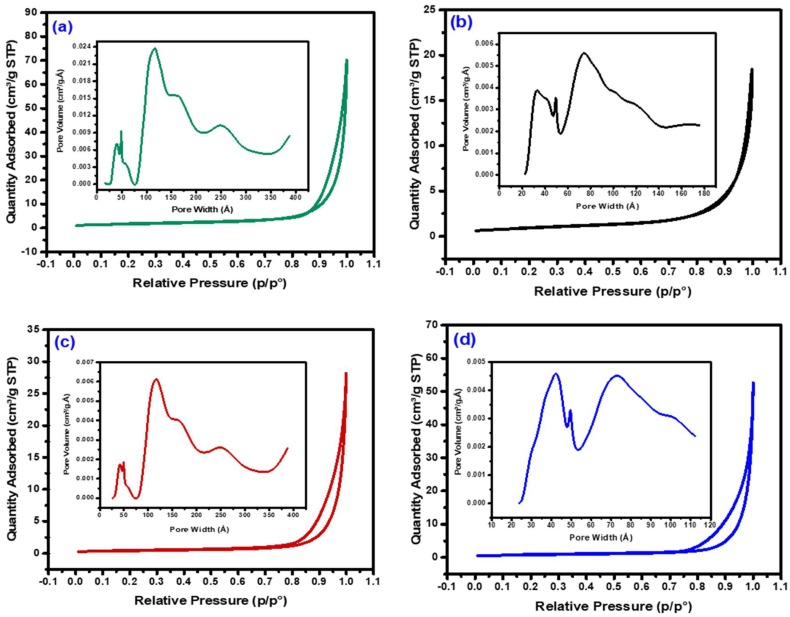
Nitrogen physisorption isotherms of the fresh samples: (**a**) Ni, (**b**) Ni-Co, (**c**) Ni-Fe, and (**d**) Ni-Cu.

**Figure 4 nanomaterials-08-01053-f004:**
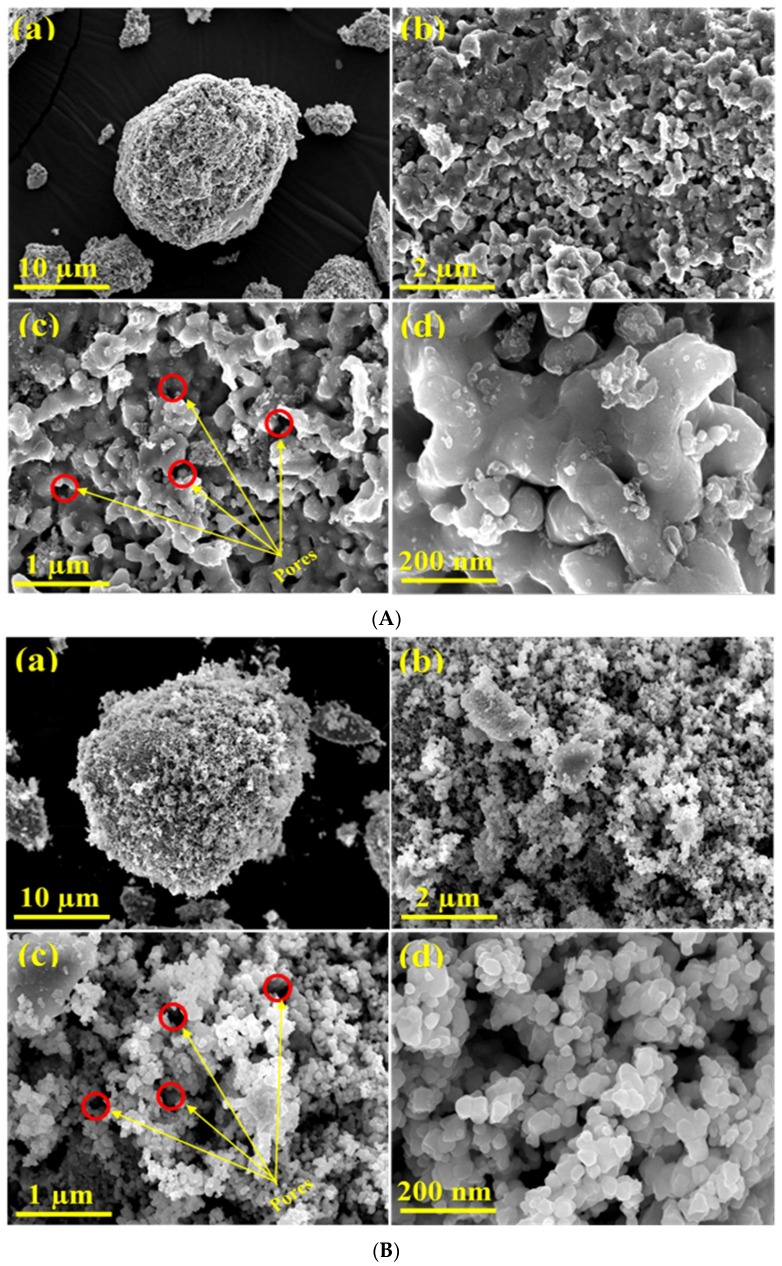
FESEM images of the fresh samples: (**A**) Ni, (**B**) Ni-Co, (**C**) Ni-Fe, and (**D**) Ni-Cu.

**Figure 5 nanomaterials-08-01053-f005:**
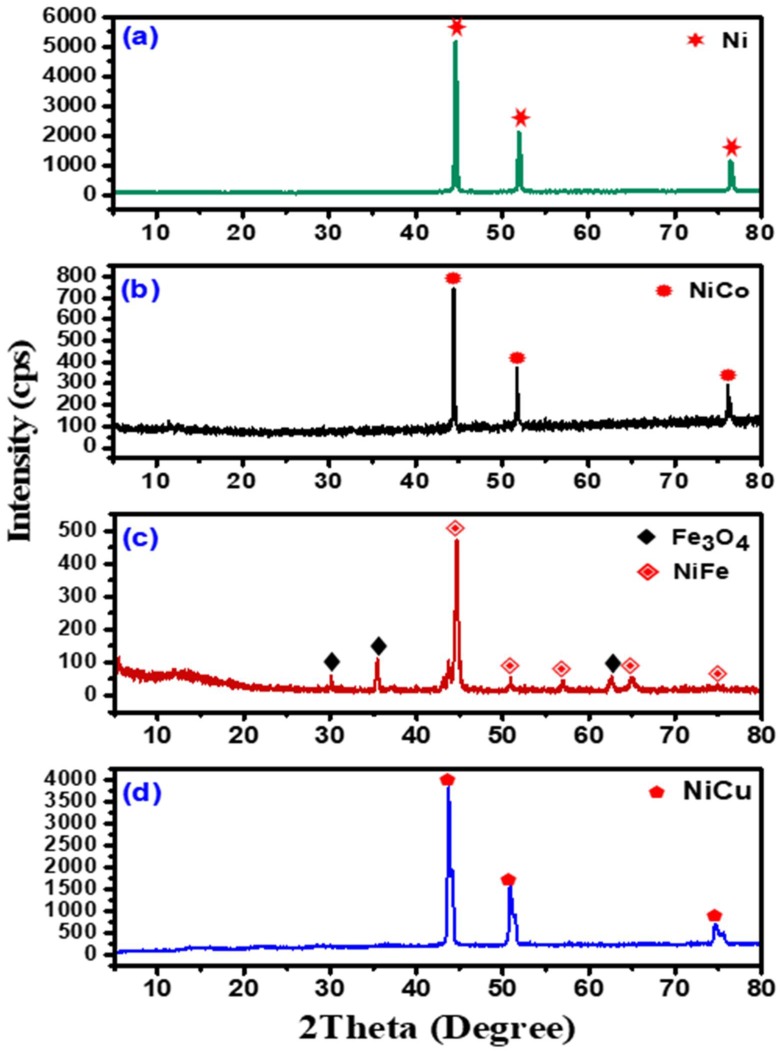
XRD patterns of the reduced samples: (**a**) Ni, (**b**) Ni-Co, (**c**) Ni-Fe, and (**d**) Ni-Cu.

**Figure 6 nanomaterials-08-01053-f006:**
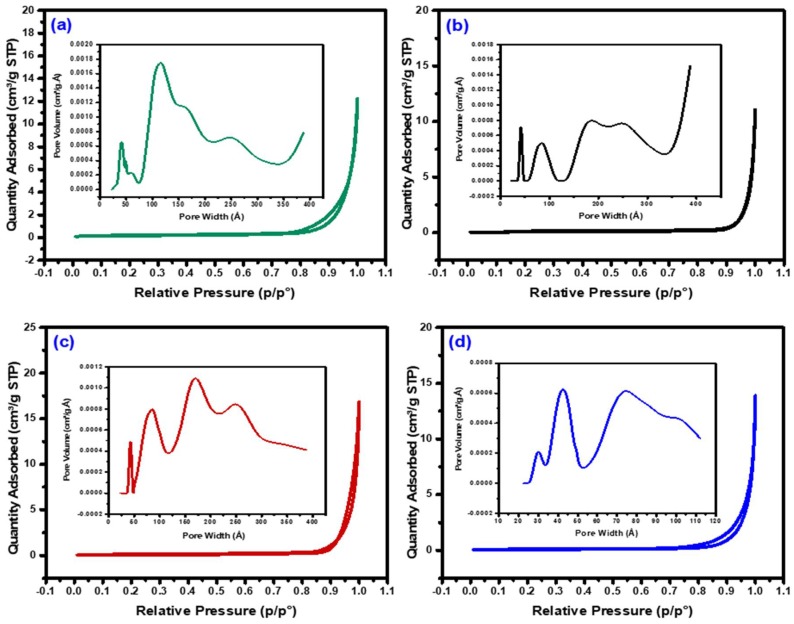
Nitrogen physisorption isotherms of the reduced samples: (**a**) Ni, (**b**) Ni-Co, (**c**) Ni-Fe, and (**d**) Ni-Cu.

**Figure 7 nanomaterials-08-01053-f007:**
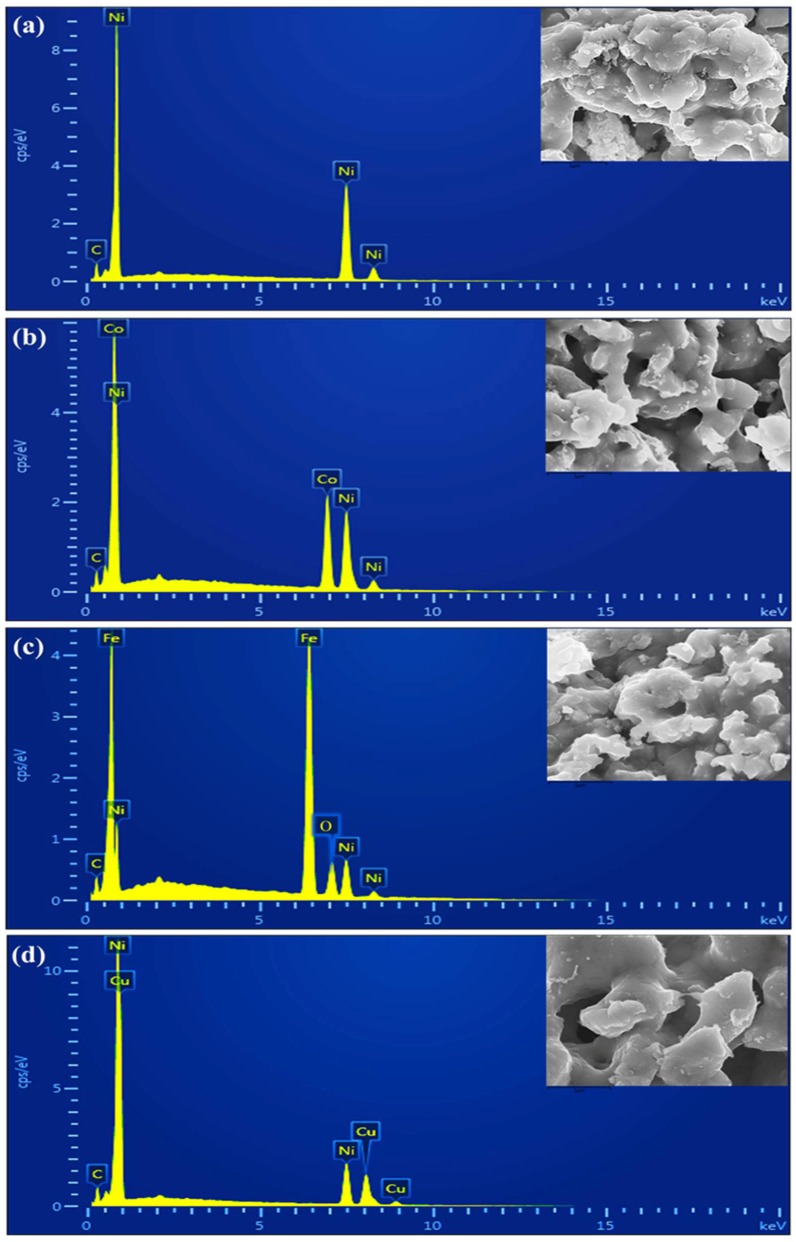
EDX spectra of the reduced samples: (**a**) Ni, (**b**) Ni-Co, (**c**) Ni-Fe, and (**d**) Ni-Cu.

**Figure 8 nanomaterials-08-01053-f008:**
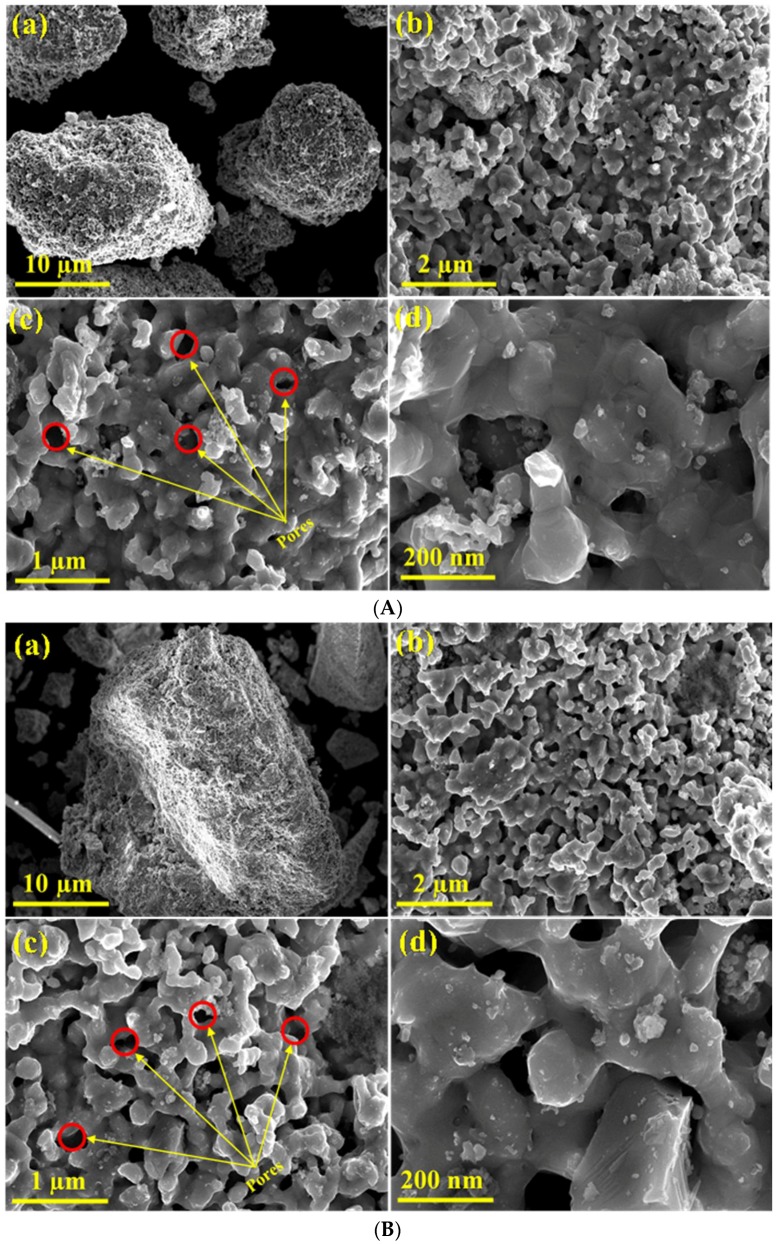
FESEM images of the reduced (**A**) Ni, (**B**) Ni-Co, (**C**) Ni-Fe, and (**D**) Ni-Cu.

**Figure 9 nanomaterials-08-01053-f009:**
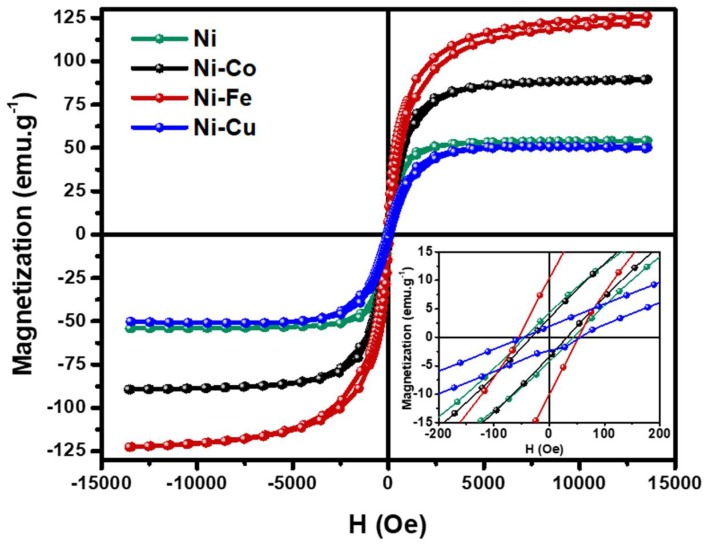
Magnetic hysteresis loops of the reduced samples ofNi, Ni-Co, Ni-Fe, and Ni-Cu. The insets show the low field detail of the magnetic hysteresis loops.

**Figure 10 nanomaterials-08-01053-f010:**
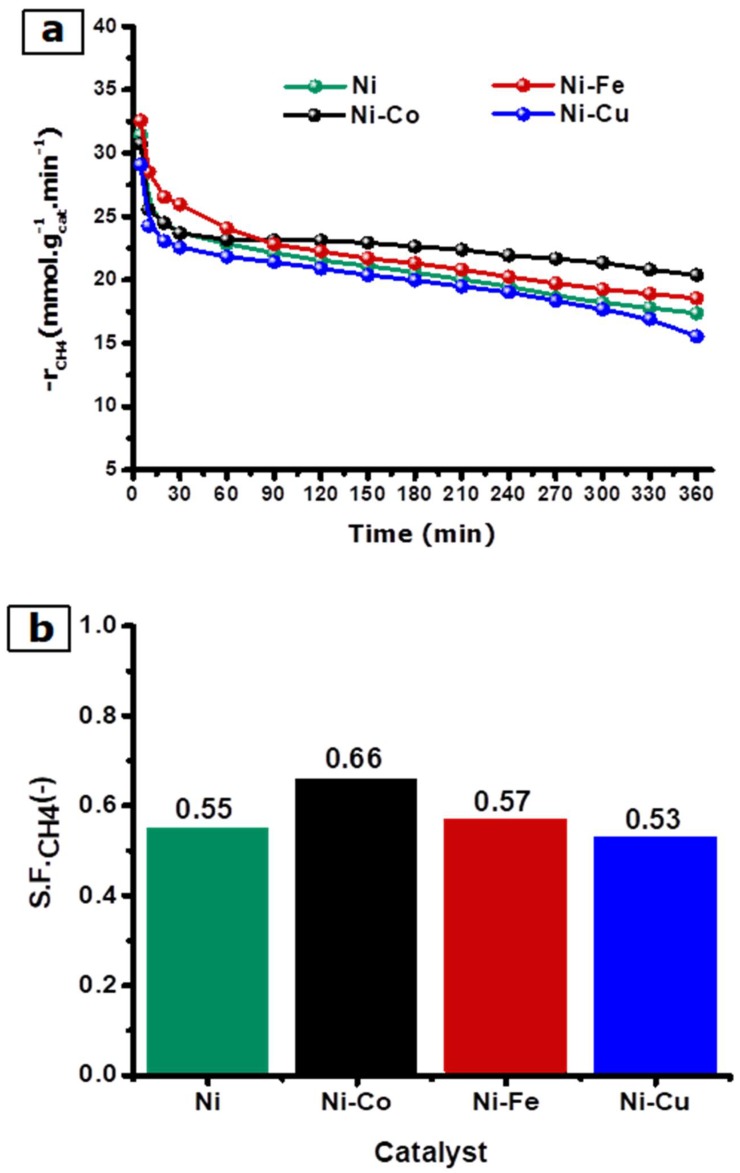
(**a**) Methane reaction rate ($-r_{{\mathrm{CH}}_{4}}$) as a function of TOS and (**b**) sustainability factors (${S.F}_{{\mathrm{CH}}_{4}}$) for Ni, Ni-Co, Ni-Fe, and Ni-Cu at 700 °C and 100 mL min^−1^.

**Figure 11 nanomaterials-08-01053-f011:**
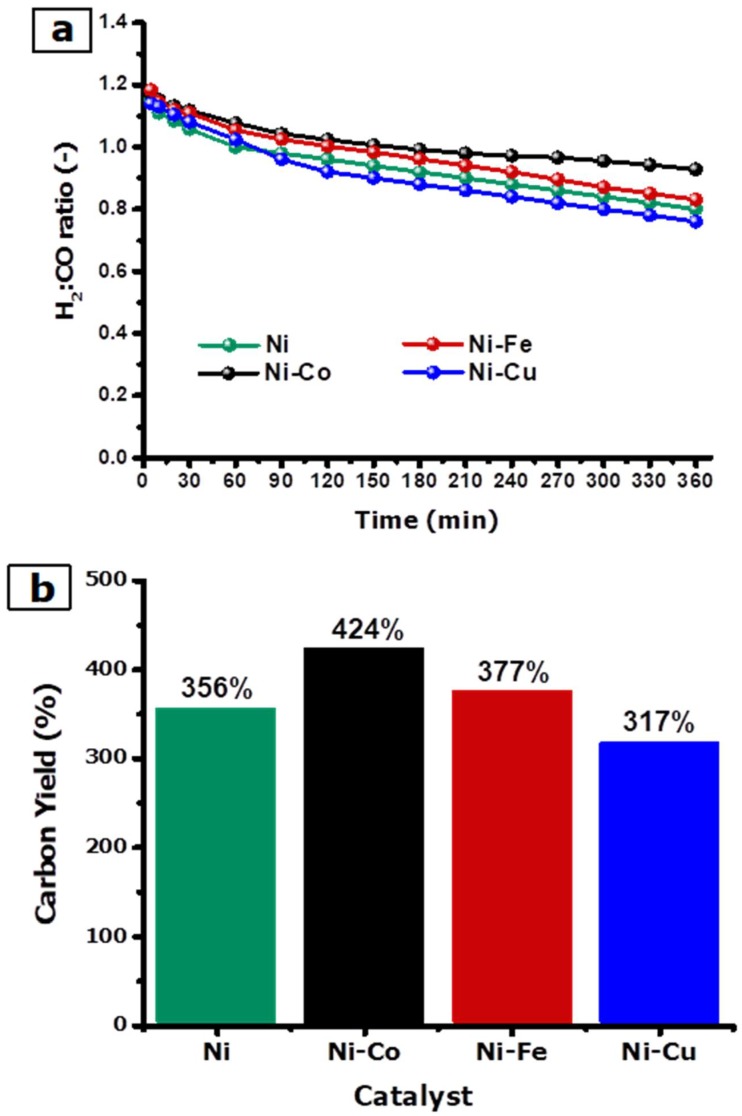
(**a**) H_2_:CO ratio as a function of TOS and (**b**) carbon yield percentage (*Y_C_*) for Ni, Ni-Co, Ni-Fe, and Ni-Cu at 700 °C and 100 mL min^−1^.

**Figure 12 nanomaterials-08-01053-f012:**
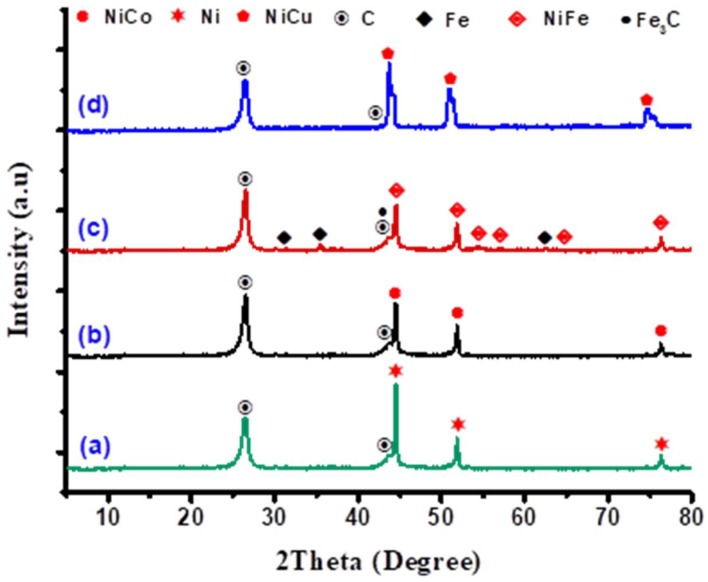
XRD patterns of the carbonous materials obtained after the reaction for (**a**) Ni, (**b**) Ni-Co, (**c**) Ni-Fe, and (**d**) Ni-Cu catalysts.

**Figure 13 nanomaterials-08-01053-f013:**
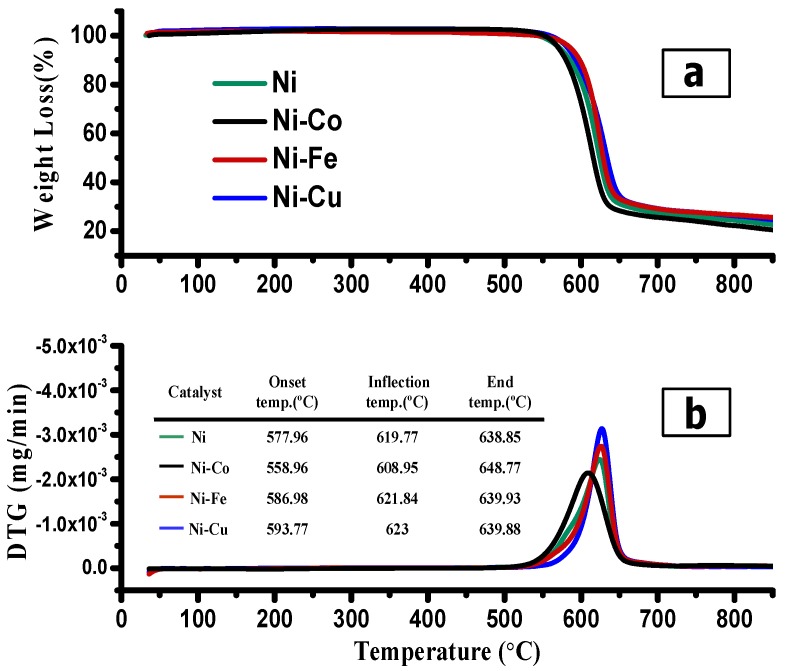
(**a**) TGA and (**b**) DTA analysis of the carbonous materials deposited over Ni, Ni-Co, Ni-Fe, and Ni-Cu catalysts.

**Figure 14 nanomaterials-08-01053-f014:**
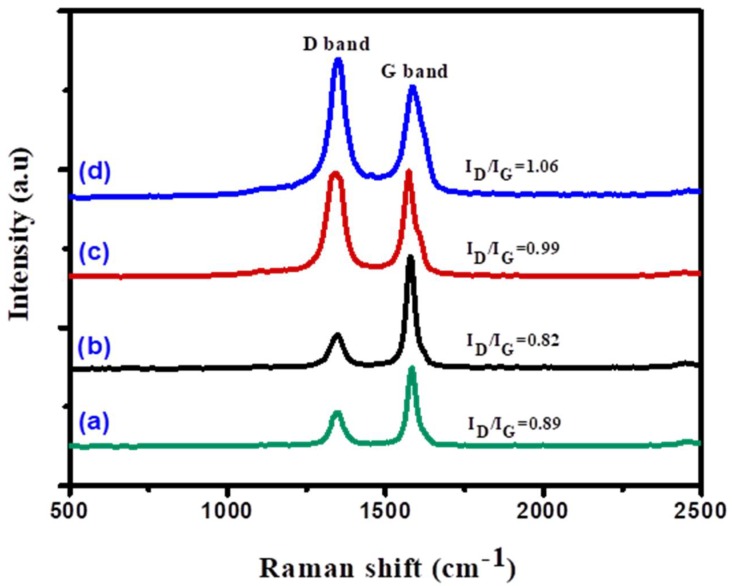
Raman spectra of the carbonous materials deposited over (**a**) Ni, (**b**) Ni-Co, (**c**) Ni-Fe, and (**d**) Ni-Cu catalysts. The morphological structures of the deposited carbon samples were described using the TEM technique, and the micrographs are presented in Figure 15 and Figure 16.

**Figure 15 nanomaterials-08-01053-f015:**
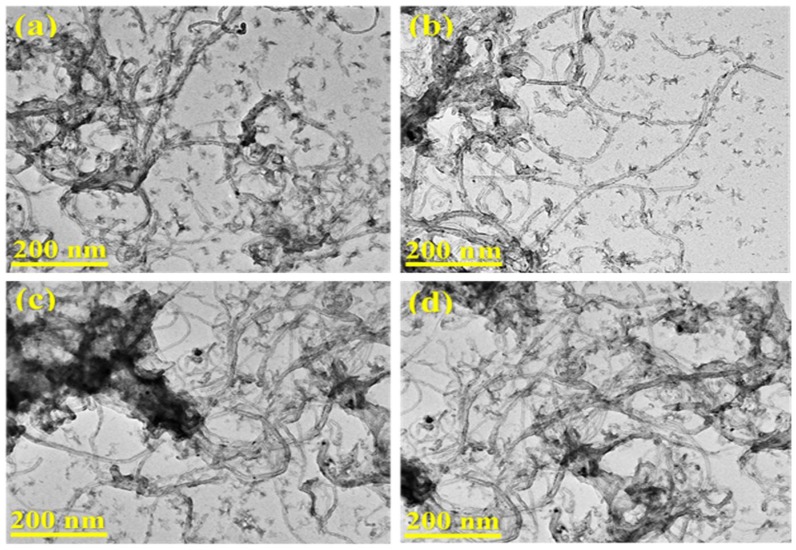
TEM images of the carbonous materials deposited over (**a**) Ni, (**b**) Ni-Co, (**c**) Ni-Fe, and (**d**) Ni-Cu catalysts.

**Figure 16 nanomaterials-08-01053-f016:**
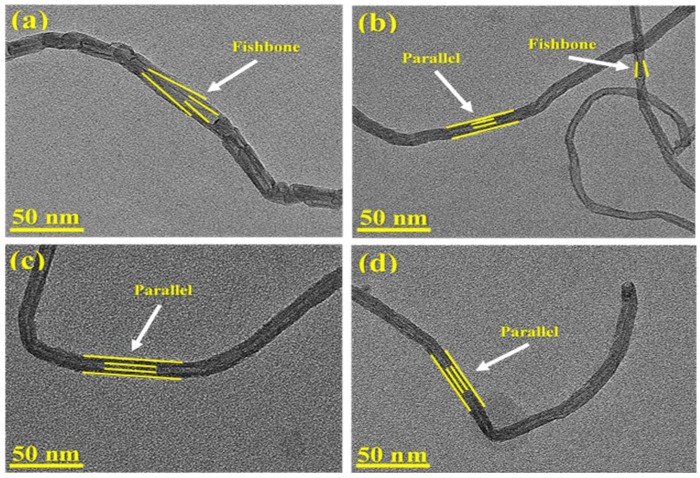
High magnification TEM images of the carbonous materials deposited over (**a**) Ni, (**b**) Ni-Co, (**c**) Ni-Fe, and (**d**) Ni-Cu catalysts.

**Table 1 nanomaterials-08-01053-t001:** Theoretical and experimental quantification of the amount of hydrogen consumed by the alloys in TPR analysis.

H_2_ Consumption (µmol g^−1^)								
**Theoretical**								
**Sample**	**NiO to Ni(0)**	**Co_3_O_4_ to CoO**	**CoO to Co(0)**	**Fe_2_O_3_ to Fe_3_O_4_**	**Fe_3_O_4_ to FeO**	**FeO to Fe(0)**	**CuO to Cu(0)**	**Total**
**Ni**	15,694	-	-	-	-	-	-	15,694
**NiCo**	6887	534	6331	-	-	-	-	13,752
**NiFe**	11,538	-	-	1799	3722	6475	-	23,534
**NiCu**	7187	-	-	-	-	-	6749	13,936
**Experimental**								
**Sample**	**Peak1 (P1)**		**Peak2 (P2)**		**Peak3 (P3)**		**Total**	
**Ni**	4591		9013		-		13,604	
**NiCo**	10,922		4211		-		15,133	
**NiFe**	1002		8379		7042		16,423	
**NiCu**	8977		1420		2318		12,715	

**Table 2 nanomaterials-08-01053-t002:** Textural properties and crystal size of the fresh oxide and bimetallic alloy catalysts determined by N_2_ adsorption and XRD.

Sample	Specific Surface Area (m^2^/g)	Pore Volume of Pores (cm^3^/g)	Pore Diameter (nm)	Crystalline Size ^a^ (nm)
**Freshly Oxides**				
**Ni**	5.8	9.8 × 10^−3^	6.7	39
**Ni-Co**	3.2	3.9 × 10^−3^	7.4	35
**Ni-Fe**	1.5	3.8 × 10^−3^	8.7	34
**Ni-Cu**	2.7	5.4 × 10^−3^	4.1	33
**Bimetallic Alloys**				
**Ni**	0.6	4.4 × 10^−3^	4.4	33
**Ni-Co**	0.2	1.2 × 10^−3^	5.9	26
**Ni-Fe**	0.3	2.8 × 10^−3^	5.8	15
**Ni-Cu**	0.3	2.2 × 10^−3^	3.8	11

^a^ Size calculated based on the XRD peak width using Scherrer’s equation.

**Table 3 nanomaterials-08-01053-t003:** The remanence (Mr), saturation magnetization (Ms), and coercivity (Hc) magnetic parameters of the alloys.

Bimetallic Alloy	Crystalline Size ^a^ (nm)	Structure (Based on XRD)	Mr (emu g^−1^)	Ms (emu g^−1^)	Hc (Oe)
**Ni**	33	fcc	4.3	54.2	43.7
**Ni-Co**	26	fcc	3.4	89.5	31.9
**Ni-Fe**	15	fcc	10.2	124.4	53.7
**Ni-Cu**	11	fcc	1.9	51.0	46.5

^a^ Size calculated based on the XRD peak width using Scherrer’s equation.

**Table 4 nanomaterials-08-01053-t004:** CH_4_ and CO_2_ conversions, syngas H_2_ and CO concentrations (*v*:*v*, dry basis), H_2_:CO ratio, and reaction rate obtained after 5 min on stream and CH_4_ sustainability factor, carbon yield, and carbon efficiency obtained after 360 min on stream at 700 °C and 100 mL min^−1^.

Catalyst	$X_{{\mathrm{CH}}_{4}}$ (%)	$X_{{\mathrm{CO}}_{2}}$(%)	H_2_ (%)	CO (%)	Ratio H_2_:CO	$-r_{{\mathrm{CH}}_{4}}$(%)	${S.F}_{{\mathrm{CH}}_{4}}$ (-)	Yc (wt.%)	$C_{E}\;\left({\mathrm{mol}}_{C}\cdot {\mathrm{mol}}_{{\mathrm{CH}}_{4}}^{-1}\times 100\right)$
**Ni**	67.19	78.61	42.03	36.40	1.15	31.42	0.55	356	28.44
**Ni-Co**	65.73	81.10	43.34	37.03	1.17	30.73	0.66	424	36.43
**Ni-Fe**	69.60	87.55	44.73	37.79	1.18	32.54	0.57	377	34.97
**Ni-Cu**	62.17	74.95	39.43	34.58	1.14	29.07	0.53	317	19.22

($X_{{\mathrm{CH}}_{4}}$) Methane conversion, ($X_{{\mathrm{CO}}_{2}}$) carbon dioxide conversion, ($-r_{{\mathrm{CH}}_{4}}$) reaction rate, (${S.F}_{{\mathrm{CH}}_{4}}$) CH_4_ sustainability factor, (H_2_) hydrogen yield, (CO) carbon monoxide yield, and H_2_:CO ratio (hydrogen yield/carbon monoxide yield).

**Table 5 nanomaterials-08-01053-t005:** Interplanar distance, carbon crystal domain size, catalyst crystal domain size, I_D_/I_G_ ratio, and carbon bio-nanofilaments diameter of the carbonous materials obtained after the reaction.

Sample	XRD			I_D_/I_G_	TEM
	d_002_ (nm)	L_C_ (nm)	d_S-cat_ (nm)		d_CNBF_ (nm)
**Ni**	0.333	6.9	35.8	0.89	42.69 ± 18.88
**Ni-Co**	0.343	5.1	32.7	0.82	13.98 ± 4.51
**Ni-Fe**	0.336	5.1	23.9	0.99	12.30 ± 3.84
**Ni-Cu**	0.334	6	24.2	1.06	11.91 ± 3.30

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
