# Peer review of "Non-Supported Nickel-Based Coral Sponge-Like Porous Magnetic Alloys for Catalytic Production of Syngas and Carbon Bio-Nanofilaments via a Biogas Decomposition Approach"

_nanomaterials, 2018, doi:10.3390/nano8121053_

Reviewer 1 Report

The authors prepared. characterized sponge-like metal alloys based on Ni and used them in the conversion of biogas into synthesis gas and carbon materials. The methods chosen provide valuable information and the comparison of different alloys in the title reaction is quite an important result.

There a few comments:

The authors should provide the name of the mysterious compound C6H8O7.H2O used as a precipitating agent 

The preparation procedure is not clear: "Firstly, the mixture of Ni(NO3)2.6H2O (0.1 mol), M (M=Co, Fe, Cu) (0.1 mol) and (NH4)2CO3 (0.2 mol) was thawed in one litter beaker containing five hundred milliliters of deionized water." Does it mean that Ni nitrate was mixed with a second metal (not salt) and ammonium carbonate? Then it is not understandable what happens next. Further: "dried for half a day" means 4 h (half a working day) or 12 h?

The alloy samples prepared must be pyrophoric. How they can be handled?

Reviewer 2 Report

The manuscript describes the production of magnetic coral sponge like porous Ni, Ni-Co, Ni-Fe and Ni-Cu bimetallic alloys via a facile double pot method. The unsupported alloys were tested as catalysts for the direct decomposition of biogas (CH4 and CO2) in order to produce syngas (H2 and CO).

Here are my comments to the authors:

- Authors should carefully review the literature and include more recent publications such as: Catalysts 2018, 8, 300; doi: 10.3390 / catal8080300, https://doi.org/10.1002/ente.201800593 or Appl. Catal. B Environ. 2017, 200, 255-264).

- Authors should withdraw equations 1-7 from the Materials and Methods section and place the meaning of these parameters in the appropriate places as for example in table 4 where these parameters are presented.

- The authors should clarify equation 7. It does not make sense.

- The authors should clarify the meaning of carbon yield and its values greater than 100. It does not make sense to do as a percentage (per 100). The authors must express the accumulated carbon determined from methane conversion in grams of carbon generated per gram of catalyst (gcarbon.gramcat-1).

- The lower carbon yield in the decomposition of synthetic biogas (CH4/CO2 mixture) using Ni-Co alloys, with respect to that of Ni alone, was already reported in previous works (Catalysts 2018, 8, 300; Environ. 2017, 200, 255-264). The authors should clarify the novelty of this work either in the abstract or in the conclusions.

- Correct "physiochemical" at line 37, and double commas at line 87. Review the entire text.

The manuscript requires major revision before publication.

Reviewer 3 Report

Authors reports on the preparation of unsupported bimetallic Ni based catalysts for the production of syngas.

Some revisions are required before its acceptance in "Nanomaterials" Journal.

- the formation of bimetallic catalysts should be confined at least with XPS analysis.

- the different reactivities should be explained taking into account the electronic properties of Ni in presence of co-metals.

- A reaction mechanism on the presence of bimetallic catalyst should be provided

Author Response

Round  2

Reviewer 1 Report

The authors took into account all the comments             

Reviewer 2 Report

I agree that the manuscript should be published at this stage.

Reviewer 3 Report

Authors successfully addressed raised points